# Assessing Whether Morphological Changes in Axillary Lymph Node Have Already Occurred Prior to Metastasis in Breast Cancer Patients by Ultrasound

**DOI:** 10.3390/medicina58111674

**Published:** 2022-11-18

**Authors:** Qiang Guo, Zhiwu Dong, Lixin Jiang, Lei Zhang, Ziyao Li, Dongmo Wang

**Affiliations:** 1Department of Ultrasound Medicine, Qingpu Branch of Zhongshan Hospital Affiliated to Fudan University, Shanghai 201700, China; 2Department of Laboratory Medicine, Jinshan Branch of Shanghai Sixth People’s Hospital Affiliated to Shanghai Jiaotong University, Shanghai 201599, China; 3Department of Ultrasound in Medicine, Renji Hospital Affiliated to Shanghai Jiaotong University, Shanghai 201599, China; 4Department of Ultrasound Medicine, the Second Affiliated Hospital of Harbin Medical University, Harbin 150086, China

**Keywords:** lymph node, ultrasonography, breast neoplasms, lymphatic metastasis

## Abstract

*Background and Objectives:* Whether the morphological changes in axillary lymph node (ALN) have occurred prior to metastasis remains unclear in breast cancer (BC) patients. The aim of this study is to investigate the influence of BC for the morphology of non-metastasis ALN (N−) and, further, to improve the performance of ultrasound (US) examination for metastasis ALN (N+). *Materials and Methods:* In this retrospective study, 653 patients with breast mass were enrolled and divided into normal group of 202 patients with benign breast tumor, N− group of 233 BC patients with negative ALN and N+ group of 218 BC patients with positive ALN. US features of ALN were evaluated and analyzed according to long (L) and short (S) diameter, the (L/S) axis ratio, cortical thickness, lymph node edge, replaced hilum and color Doppler flow imaging (CDFI). *Results:* ALN US features of short diameter, replaced hilum, cortical thickness and CDFI have significant statistical differences in N− group comparing with normal group and N+ group, respectively (*p* < 0.05). *Conclusions:* Therefore, BC can affect ALN and lead to US morphological changes whether lymph node metastasis is present, which reduces the sensitivity of axillary US. The combination of US and other examination methods should be applied to improve the diagnostic performance of N+.

## 1. Introduction

The status of axillary lymph node (ALN) is an important prognostic factor and has a crucial impact on both surgical and therapeutic management in breast cancer (BC) [1,2]. Therefore, correct evaluation for malignant or benign tendency of ALN plays a great role in determining the risk of recurrence and survival in BC patients. Nowadays, axillary lymph node dissection (ALND) is changing to sentinel lymph node dissection (SLND) in evaluating the status of ALN for less invasive and fewer complications in BC patient [3]. However, SLND still causes a certain degree of false negative rate and some complications such as shoulder and back pain, arm numbness, weakened shoulder and reduced arm strength [4,5]. Therefore, some noninvasive examination methods, such as ultrasound (US), magnetic resonance imaging (MRI) and contrast enhanced computerized tomography (CT), have been widely applied to evaluate the status of ALN.

With the development of ultrasound technology, axillary ultrasound (AUS) examination has been proved to be an effective method in assessing N+ and the clinical staging of BC [6,7,8]. Some useful US features, such as longitudinal and transverse (L/T) ratio, cortical thickness and color Doppler flow imaging (CDFI), have been widely applied to assess ALN and provided useful reference for the tendency of N+ [1,9]. In recent years, artificial intelligence (AI) and radiomics have increasingly been used and achieved significantly better performance in predicting N+ in BC [10]. Fanizzi, A. et al. applied machine learning algorithms to estimate N+ with clinical and pathological data, and reached a better sensitivity value of 72% [11]. Radiomic features with lots of information extracted from primary BC ultrasound images have also been used to predict the ALN status, which represented a promising non-invasive predictive method and concluded some better prediction models [12,13,14].

However, the metastatic process of BC cells is more complicated due to various mechanisms and interaction of protein and acceptor, and even unclear in some of them up to today. Furthermore, BC cells can also produce different kinds of hormones to participate in the metastatic process, such as vascular endothelial growth factor (VEGF) that can promote formation of tumor’s vessel and lymphatics [15]. In view of this complicated process of BC metastasis, the morphological changes of surrounding tissue and ALN have probably occurred before metastasis of carcinoma cells. Therefore, it is debatable that evaluation of the N+ of BC is performed only according to US morphological changes.

Although, AUS shows a great performance for diagnosis of N+, there are still high false positive rates in the diagnostic process that makes some patients suffer from great pain of ALND [16,17]. Some studies demonstrated that AUS had a great sensitivity and specificity for predicting the tendency of N+ [18,19,20], while there were other studies that showed low sensitivity and specificity, or even no predictive value [21,22]. Therefore, it is worth thinking deeply about the reason of contradiction and need to be explained in further research.

It remains unclear whether the morphological characteristics of ALN have changed before metastasis up to now, which may affect the performance of AUS. Therefore, we aim to explore the influence of BC for ALN by evaluating the US morphological changes of ALN in different status of metastasis to improve the performance AUS.

## 2. Materials and Methods

### 2.1. Ethical Approval

All procedures of this study were in accordance with the ethical standard of 1964 Helsinki declaration. The ethics committee of the Second Affiliated Hospital of Harbin Medical University approved the research process. Only verbal informed consent of using their data from all the patients was required owing to the retrospective study design, noninvasive nature and use of anonymous data.

### 2.2. Patients

From January 2015 to May 2019, 653 patients with breast mass were enrolled in this retrospective study and divided into normal group of 202 patients with benign breast tumor, N− group of 233 BC patients without N+ and metastasis group of 218 BC patients with N+ according to postoperative pathological results. ALND or lymph node biopsy and pathological examination were carried out for the BC patients according to the preoperative US localization of ALN. The inclusion criteria were as follows: (1) having complete results of pathological diagnosis and immunohistochemical molecular markers in BC patients; (2) having no change in long-term observation for at least two years in patients without pathological results; (3) having complete results of axillary US examinations and localization for suspicious ALN. Patients who have received any treatments before the operations were excluded.

### 2.3. Ultrasound Examination

Two sonographers who had ten years’ experience in breast and axillary US performed the preoperative axillary ultrasonic examination and lymph node localization for all the patients by HITACHI Vision 900 system (Hitachi Medical System, Tokyo, Japan) or S2000 (Siemens Medical Solutions, Mountain View, CA, USA) with a linear-array transducer of 5–12 MHz. The patient lay in dorsal decubitus position with the hand placed on the head, and a continuous scanning of the axilla was completed with the aim of identifying the suspicious lymph nodes.

In each case, ultrasound features of the suspicious lymph node were analyzed and recorded by conventional ultrasound images (B-mode images) according to long (L) diameter, short (S) diameter, L/T axis ratio, cortical thickness, replaced hilum (yes or no) and edges (regular or irregular). The features of CDFI were evaluated and divided into four grades of 0, 1, 2 and 3 using the Adler’s grading method [23]. We classified the axillary lymph nodes as positive results according to the follow ultrasonic imaging criteria: (1) short diameter ≥ 7 mm, (2) (L/T) axis ratio ≥ 2 mm, (3) eccentric or concentric cortical thickness ≥ 3 mm, (4) having replaced hilum, (5) having irregular edges morphology and (6) 2 to 3 grade blood flow [24,25]. Figure 1 showed the US features of ALN.

### 2.4. Histological Examination

Histological diagnosis was performed to identify the benign and malignant for surgical specimens of breast tumor using routine methods of hematoxylin-eosin (HE) stain, formalin-fixed and paraffin-embedded material. For the malignant tumor, the further pathological examination and immunohistochemical method were routinely performed for tumor size, type of pathology, nuclear grades, histologic grades, venous invasion, ER, PR and HER-2. Sentinel lymph node biopsy (SLNB) or ALND was performed by a doctor with five years experienced breast surgeons. ALND was completed for the patients with metastatic lymph nodes by fine-needle aspiration (FNA), While those with negative FNA results received SLNB. The sentinel lymph node status was identified by frozen pathologic testing to further decide ALND. The number and stage of ALN were described according to the seventh edition of the American Joint Committee on Cancer (Chicago, IL, USA) Breast Cancer Staging Manual.

### 2.5. Statistical Analyses

The descriptive statistics were divided into continuous and categorical variables. Continuous data were expressed as the mean ± standard error and categorical data were expressed as a percentage. Chi-squared tests was performed for categorical variables to compare the US characteristics of ALN among N−, metastasis and normal groups. Multivariate logistic regression models were built by a stepwise regression method for searching for meaningful variables. Inter-observer agreement was assessed with the Cohen’s kappa statistics. *p* value < 0.05 was considered statistically significant. The statistical analysis was completed through SPSS 18.0 (Chicago, IL, USA).

## 3. Results

### 3.1. Clinical Characteristics

There were 233 BC patients without N+ as N− group, 218 BC patients with N+ as metastasis group and 202 patients with benign breast mass as benign group. The mean age (47 ± 19.7 years, 49 ± 18.3 years and 46 ± 17.8 years) and the mean size of mass (21 ± 11.7 mm, 23 ± 12.5 mm and 20 ± 10.9 mm) showed no significant differences among the N− group, N+ group and benign group (*p* > 0.05). Again, we found no significant differences among the three groups in terms of body mass index (BMI) (<25 kg/m^2^, ≥25 kg/m^2^) and comorbidities (hypertension, diabetes, cardiovascular disease, and others) (*p* > 0.05). The histologic results showed that 202 cases were confirmed as benign masses including fibroma (156 cases) and lobular hyperplasia (46 cases), and 451 cases were confirmed as malignant tumor including invasive ductal carcinomas (360 cases), invasive lobular carcinomas (29 cases) and other subtypes (62 cases). No significant difference was observed in those parameters of histological type, histological grade, estrogen receptor (ER), progesterone receptor (PR) and epidermal growth factor receptor 2 (HER-2) comparing N− group with N+ group in BC patients (*p* > 0.05), and the results were listed in Table 1.

### 3.2. Correlational Analyses of Ultrasound Features of Axillary Lymph Node

The number of ALN with short diameter ≥ 7 mm was larger in the N− group (146 out of 233, 62.7%) compared with the normal group (98 out of 202, 49%, *p* = 0.005), while it was smaller compared with the N+ group (164 out of 218, 75.2%, *p* = 0.004). In terms of the L/S ratio of ALN, no significant statistical differences were observed between the N− group and normal group, and the N− group and N+ group (*p* > 0.05). The number of ALN with cortical thickness ≥ 3 mm was larger in N− group (108 out of 233, 46.4%) compared with the normal group (71 out of 202, 35.1%, *p* = 0.025), while it was smaller compared with the N+ group (124 out of 218, 56.9%, *p* = 0.018). Replaced hilum of ALN is more likely to occur in the patients of N− group (46 out of 233, 19.7%) than the normal group (11 out of 202, 5.4%, *p* < 0.001), while it is not easier to present compared with the N+ group (112 out of 218, 51.4%, *p* < 0.001). With respect to the regular or irregular edges, there were no significant statistical differences among the three groups of N− group, normal group and N+ group (*p* > 0.05). Compared with the normal group (68 out of 202, 33.7%), the CDFI grade of ALN is higher in the N− group (104 out of 233, 44.6%, *p* = 0.024), while the CDFI grade is lower than the N+ group (118 out of 218, 54.1%, *p* = 0.048). The US presentation of ALN and chi-square test results were listed in Table 2 and Figure 2.

### 3.3. Multiple Logistic Regression Analysis of Axillary Lymph Node in Different Status

Table 3 shows the results of the multivariate logistic regression that was used to more rigorously select the variables of US features of ALN status. There were three US variables as independently predictive factors including short diameter (β = 0.693; OR, 1.965; 95% CI, 0.796–3.935; *p* = 0.114), cortical thickness (β = 0.954; OR, 2.133; 95% CI, 1.635–3.164; *p* < 0.001) and CDFI grade (β = 1.196; OR, 2.923; 95% CI, 1.853–6.773; *p* = 0.001) between the N− group and normal group. A Receiver operating characteristic curve was drawn, and the area under the curve was 0.692. Furthermore, there were also three US variables as independently predictive factor including short diameter (β = 0.818; OR, 2.426; 95% CI, 1.006–3.532; *p* < 0.001), cortical thickness (β = 1.012; OR, 2.661; 95% CI, 1.635–3.164; *p* = 0.053) and replaced hilum (β = −2.776; OR, 0.058; 95% CI, 0.031–0.205; *p* < 0.001) between the N− group and N+ group. A Receiver operating characteristic curve was drawn, and the area under the curve was 0.670 (Figure 3).

### 3.4. Observer Agreement

Consistency analysis of interpretations for the US features of ALN between two sonographers were performed by Cohen’s kappa statistics. The better agreement of inter-observer was shown, the kappa value of every US feature ranges from 0.62 to 0.79.

## 4. Discussion

Axillary US has been extensively applied to evaluation for N+ in the patients with BC, which provided clinically useful information to guide surgical decision-making [6,25]. However, various studies have also pointed out that preoperative US evaluation of axillary in early-stage breast cancer was not recommended due to its low sensitivity [26,27,28]. Therefore, there is an urgent need to analyze the controversial problem in term of the performance of axillary US for lymph node metastasis to further improve the application of US. Interestingly, we concluded that the US morphological structures of ALN have changed in BC patients with N−, which may help to explain the low sensitivity of US for the N+. Therefore, the study conclusion would improve identification capability of US for N+ and increase the accuracy of FNA that was are required before ALND according to the guidelines of Society of Clinical Oncology (ASCO) and the National Comprehensive Cancer Network (NCCN).

US features of short diameter, L/T axis ratio, cortical thickness, lymph node edges, replaced hilum and the CDFI were constantly used as evaluative criteria to diagnose the metastasis of ALN [24,25]. In this study, the significant differences were observed in the three US features of short diameter, cortical thickness and CDFI in the N− group compared with the normal group. Meanwhile, between the N− group and N+ group, three US features of short diameter, cortical thickness and replaced hilum showed the significant differences. Therefore, our research results showed that the changes in US features of short diameter, cortical thickness, replaced hilum and CDFI have occurred, even if the ALN is negative in BC patients. Therefore, this was the first study to take ALN features of BC patients with N− as a primary issue to study.

To our knowledge, ALN enlargement and morphological change in BC patient are not always caused by metastasis even in the ipsilateral axillary nodes, which may be the mainly reason of low sensitivity of axillary US for N+. BC cells can release vascular endothelial growth factor (VEGF) that is a multi-functional cytokine as a major angiogenic factor to promote the formation of tumor vessels and help tumor obtaining nutrients needed for mass reproduction [29]. VEGF can more easily spread to ALN through lymphatic vascular of breast, which improve the formation of lymph node vessels and lead to lymph node hyperplasia before BC metastasis. Additionally, BC patients have high-level nitric oxide (NO) that is a potent biological molecule participating in the multi-step process of carcinogenesis, including induction of apoptosis and promotion of angiogenesis [30,31]. Inducible NO synthase can be found in peritumoral lymph nodes by immunocytochemical method [32], which proved that the NO participates in the process of ALN hyperplasia in BC patients. Furthermore, Interleukin (IL)-18 is a BC related anti-tumor factor that is mediated by induction of apoptosis and inhibition of angiogenesis [33], which can affect the morphological structure of lymph node. Therefore, the N− ALN has been affected by the tumor-associated factors and the morphological features have changed before metastasis of BC cells.

Axillary lymph tissue is more vulnerable to multiple factors and invasion of BC cells. Based on the current dogma, metastatic predisposition may be attributed to inherent molecular differences in tumor cells and surroundings of the vasculature, connective tissue and immune cells [34]. With tumor progression, tumor tissue and distant organs can secrete soluble cytokines to alter the ability of invasion and metastasis by the contact of immune cells and tumor cells. According to the pre-metastasis microenvironment hypothesis proposed by Kaplan, a large number of tumor-acclimated cells and tumor-derived molecules precede the entry of tumor cells into the organ, providing soil and indicating the direction for tumor metastasis [35]. Therefore, the process of tumor immune regulation changes the morphological structures of ALN prior to metastasis, which is consistent with our results by US in BC patients.

There were some limitations in this study, which are as follows: firstly, retrospective and single-centered research easily leads to unavoidable bias; secondly, the small sample size should be further improved; thirdly, there was a little risk that preoperative US localization may be not consistent with postoperative pathology results; fourthly, the US as a highly operator-dependent technology performed by different sonographers can influence the results of image analysis; finally, and perhaps most importantly, AI and radiomics should be performed to achieve high performance, which have been successfully applied in clinical medicine [36,37]. Further work in the area will be needed to overcome these limitations.

## 5. Conclusions

This study is the first to conclude the most interesting conclusion that the morphological features of ALN on US have changed in the BC patients with N−, which could lead to the low sensitivity of axillary US. To further improve the diagnostic performance of US for N+, a combination of multiple examination methods, such as contrast-enhanced MRI, PET-CT and AI, should be applied to the diagnostics of N+.

## Figures and Tables

**Figure 1 medicina-58-01674-f001:**
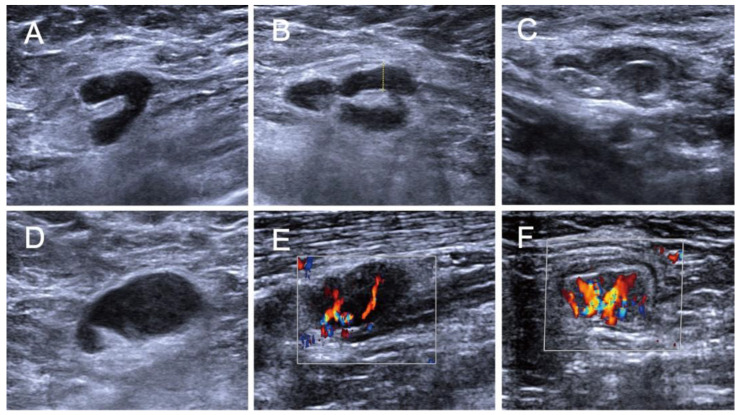
(**A**) The measuring approach of short diameter is showed in a short-axis cross-sectional US image of ALN by the white solid line. (**B**) The cortical thickness is obtained in a long-axis cross-sectional US image of ALN by the yellow dotted line. (**C**) US features of partly replaced hilum and irregular edges show in the US image of ALN. (**D**) The US image shows a morphological disorder of ALN with completely replaced hilum and irregular edges. (**E**) The US image shows a morphological disorder with completely replaced hilum and CDFI grade 2. (**F**) The US image shows the high blood perfusion into ALN and CDFI grade 3.

**Figure 2 medicina-58-01674-f002:**
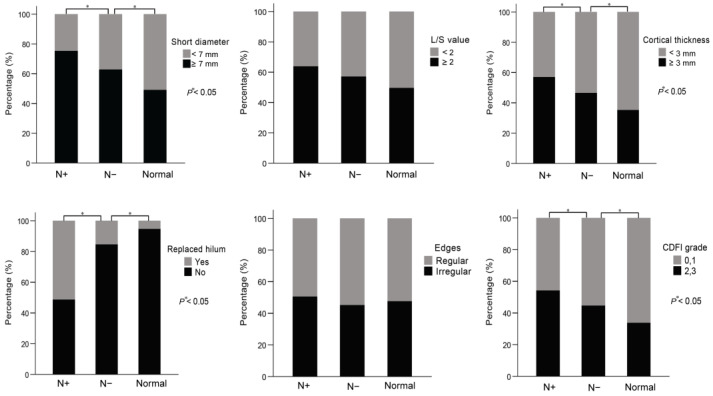
The six the ultrasound features of ALN including short diameter, L/S value, cortical thickness, replaced hilum, edges and CDFI were compared among the three groups of N+ group, N− group and normal group. The significant differences were observed in the four US features of short diameter, cortical thickness, replaced hilum and CDFI in the N− group compared with the normal group and N+ group by chi-squared tests. The * represents a statistically significant difference.

**Figure 3 medicina-58-01674-f003:**
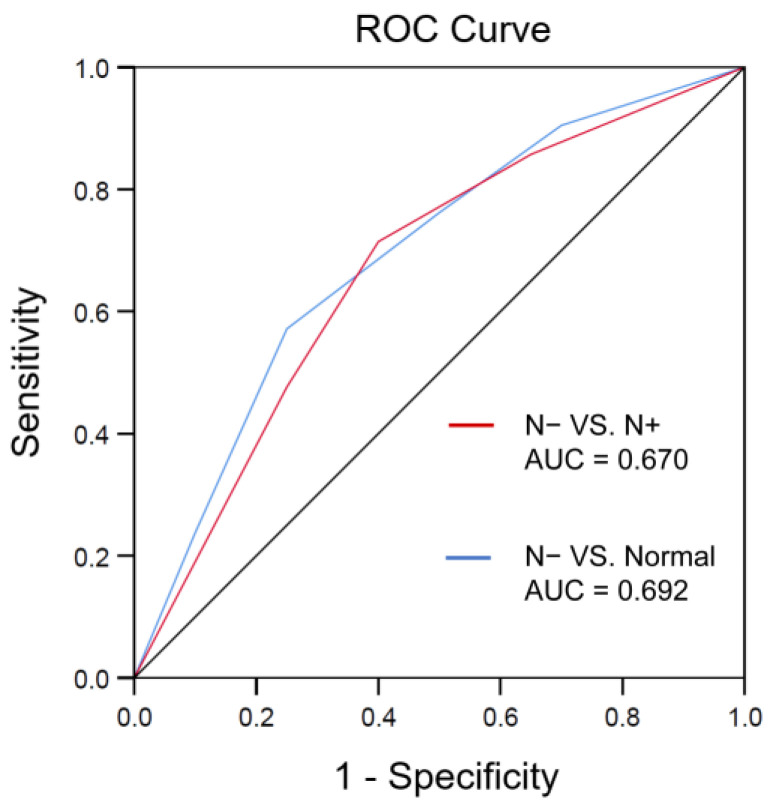
Receiver operating characteristic curve for the differentiation capacity of the multiple regression model between the N− group and N+ group in red line, and between the N− group and normal group in blue line. AUC indicates area under the curve.

**Table 1 medicina-58-01674-t001:** Patient demographics and clinical-pathological features.

Features	Breast Cancer		Benign Mass	*p* Value
	N−	N+		
No. of patients	233	218	202	
Age	47 ± 19.7 years	49 ± 18.3 years	46 ± 17.8 years	0.141
Size of tumor	21 ± 11.7 mm	23 ± 12.5 mm	20 ± 10.9 mm	0.078
BMI (kg/m^2^)				
<25	145	151	132	0.290
≥25	88	67	70	
Comorbidities				
Hypertension	31	34	22	0.567
Diabetes	26	17	19	
CVD	19	21	16	
No or Others	157	146	145	
Histological type				
IDC	126	114		0.707
Others	107	104		
Histological grade				
Grade 1	46	32		0.112
Grade 2	124	109		
Grade 3	63	77		
ER				
Positive	69	72		0.477
Negative	164	146		
PR				
Positive	175	146		0.062
Negative	58	72		
HER-2				
Positive	80	76		0.921
Negative	153	142		

Abbreviations: N− = negative ALN; N+ = positive ALN; BMI: body mass index; CVD = cardiovascular disease; IDC = invasive ductal carcinomas; ER, estrogen receptor; PR, progesterone receptor; HER-2, epidermal growth factor receptor 2.

**Table 2 medicina-58-01674-t002:** Comparing ultrasound features of axillary lymph node in different status.

Features	N− (*n* = 233)	N+ (*n* = 218)	Normal (*n* = 202)	χ^2^ *	*p* *	χ^2^ **	*p* **	χ^2^ ***	*p* ***
Short diameter									
<7	87 (37.3%)	54 (24.8%)	103 (51.0%)	8.279	0.004	11.323	0.005	36.002	<0.001
≥7	146 (63.7%)	164 (75.2%)	98 (49.0%)						
L/S									
<2	100 (42.9%)	79 (36.2%)	102 (50.5%)	1.868	0.172	2.656	0.103	8.69	0.003
≥2	133 (57.1%)	139 (63.8%)	100 (49.5%)						
Cortical thickness									
<3	125 (53.6%)	94 (43.1%)	131 (64.9%)	4.998	0.025	5.608	0.018	19.909	<0.001
≥3	108 (46.4%)	124 (56.9%)	71 (35.1%)						
Replaced hilum									
Yes	36 (15.5%)	112 (51.4%)	11 (5.4%)	1.469	<0.001	10.67	<0.001	19.25	<0.001
No	197 (84.5%)	106 (48.6%)	191 (94.6%)						
Edges									
Regular	128 (54.9%)	108 (49.5%)	106 (52.5%)	1.314	0.259	0.264	0.631	0.361	0.548
Irregular	105 (45.1%)	110 (50.5%)	96 (47.5%)						
CDFI grade									
0,1	129 (55.4%)	100 (45.9%)	134 (66.3%)	4.061	0.048	5.448	0.024	17.797	<0.001
2,3	104 (44.6%)	118 (54.1%)	68 (33.7%)						

Note: N− = negative ALN; N+ = positive ALN; L/S means the ratio of long axis and short axis of lymph node; * N− vs. N+; ** N− vs. Normal; *** N+ vs. Normal.

**Table 3 medicina-58-01674-t003:** Multiple logistic regression analysis of axillary lymph node in different status.

Intercept and Variable	β	Odds Ratio (95% CI)	*p*
N− vs. Normal			
Intercept	3.244		0.026
Short diameter	0.693	1.965 (0.796 to 3.935)	0.114
Cortical thickness	0.954	2.133 (0.893 to 3.395)	<0.001
CDFI grade	1.196	2.923 (1.853 to 6.773)	0.001
N− vs. N+			
Intercept	2.748		0.021
Short diameter	0.818	2.426 (1.006 to 3.532)	<0.001
Cortical thickness	1.012	2.661 (0.893 to 3.395)	0.053
Replaced hilum	−2.776	0.058 (0.031 to 0.205)	<0.001

## Data Availability

The data presented in this study are available on request from the corresponding author. The data are not publicly available due to privacy or ethical restrictions.

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
