# Peer review of "Assessing Whether Morphological Changes in Axillary Lymph Node Have Already Occurred Prior to Metastasis in Breast Cancer Patients by Ultrasound"

_medicina, 2022, doi:10.3390/medicina58111674_

Round 1
Reviewer 1 Report
Although the issue is key in clinical practice and the hypothesis intriguing, your retrospective findings do not support your conclusions "this study is the first time to conclude the most interesting conclusion that the morphological features of ALN on US have changed before metastasis of BC...". Rather than "before", it should be used "regardless of", which confirms the low predictivity rates of ultrasound in this setting.
Reviewer 2 Report
Authors can find my comments and suggestions in the attached file.

Reviewer 3 Report
Congratulations to the authors for their interesting retrospective study
This study needs several changes before admitting for publication
1- Table 1. The authors compare pathological features between metastasis group and non-metastasis., The only clinical information is age. There is a lack of information about important factors such is BMI and comorbidities that can explain different findings.
2- Table 1 There’s a lack of clinical information on the normal group. This group should be included on the table I
3- Results. A multivariable study searching for confusion variables is needed.
4- Results The results of a single variable (more or less than 7 mm, cortical thickness) had low accuracy. I suggest combinations of variables (For example, more than 7 mm and cortical think superior of 3 mm).
5- Discussion. International guidelines (NCCN, ASCO guidelines) recommends FNA of suspicious lymph nodes before performing an all lymph node dissection (ALND). The authors explain a different approachwith direct ALND without FNA confirmation. These differences should be explained in the discussion.
6- Conclusion Limitations should be explained in the discussion before be included in the conclusion section.
7- Conclusion: Data are not enough to conclude “that the morphological features of ALN on US have changed before metastasis of BC”.
8- I suggest referring to positive ALN or N+ instead of metastasic ALN in order to avoid confusions.
9- ALN may be confusing with ALND (all lymph node dissection). Please, consider some other acronym
Round 2
Reviewer 1 Report
Additional text editing is needed
Author Response
Additional text editing is needed
REPLY: We are so sorry for our poor language expression in our manuscript. Thanks for your valuable comments. Following your suggestion, we have carefully rewritten the sentence and marked in blue font.
Reviewer 3 Report
I really appreciate authors effort to improve the quality of the article.
showing the normal group characteristics the study has a stronger design who allows a better interpretation of the results
However the article still being hard to understand and several changes must be introduced.
The authors still referring in table 2 as Non-metastasic and Metastasis. By contrast, in introduction they refer N+ and N-. From my point of view this is confusing.
Discussion: The authors made several changes induced by reviewers. This section is now more confusing and extensive editing of English language and style is is required.
Conclusions are not fully supported by findings.
